# Impact of Through-Hole Defects on the Electro-Explosive Properties of Exploding Foil Transducers

**DOI:** 10.3390/mi14081499

**Published:** 2023-07-26

**Authors:** Kexuan Wang, Jiangxu Wang, Xinyu Li, Dangjuan Li, Junxia Cheng, Jia Wang, Shenjiang Wu

**Affiliations:** 1Shaanxi Applied Physics and Chemistry Research Institute, Xi’an 710061, China; 2School of Optoelectronic Engineering, Xi’an Technological University, Xi’an 710021, China; 3Xi’an North Qinghua Electromechanical Co., Ltd., Xi’an 710025, China

**Keywords:** metal bridge foil, through-hole defects, explosion time, transducer efficiency

## Abstract

This study examines the impact of surface defects on the electro-explosive properties of metal explosive foil transducers. Specifically, it focuses on the effects of defects in the bridge foil and their influence on the electrical explosion time and transduction efficiency. To analyze these effects, a current-voltage simulation model is developed to simulate the behavior of a defective bridge foil. The simulation results are validated through experimental current-voltage measurements at both ends of the bridge area. The findings reveal that the presence of through-hole defects on the surface of the bridge foil leads to an advancement in the electrical explosion time and a reduction in the transduction efficiency of the bridge foil. A performance comparison is made between the defective bridge foil and a defect-free copper foil. As observed, a through-hole defect with a radius of 20 μm results in a 1 ns advance in the blast time and a 1.52% decrease in energy conversion efficiency. Similarly, a through-hole defect with a radius of 50 μm causes a 51 ns advancement in the blast time and a 13.96% reduction in the energy conversion efficiency. These findings underscore the detrimental effects of surface defects on the electro-explosive properties, emphasizing the importance of minimizing defects to enhance their performance.

## 1. Introduction

Pyrotechnics are ignition devices that utilize control information to convert initial energy into detonation energy [1]. They are capable of conducting and amplifying combustion or explosion energy in a specific manner to achieve work, detonation, and ignition. Among the various types of pyrotechnics, the exploding foil initiator (EFI), also known as the impact piece detonator, was initially developed by Stroud et al. at Moore National Laboratory, Lawrence Livermore National Laboratory (LLNL), USA [2]. Due to its high reliability, safety, high detonation rate, and resistance to interference [3], the EFI has gained increasing usage in engineering applications.

The transducer is a crucial component of the EFI system as it converts electrical energy into energy that is required for explosion. The structure, size, materials, and other factors of the transducer influence the energy utilization of the explosive foil detonation system, plasma generation, flying sheet speed, and other parameters [4,5,6]. Consequently, the electrical explosion performance of the bridge foil transducer is an important area of research both domestically and internationally. In the process of pyrotechnic detonation, the metal bridge foil and explosive column of the transducer are not in direct contact. The bridge foil undergoes electrical explosion only when subjected to a specific pulse of high current, thereby driving the shearing of the flying piece to achieve ignition or detonation [7].

Magnetron sputtering and photolithography are commonly employed processes for fabricating blast foil transducer elements [8,9,10]. During the fabrication of metal bridge foils, various defects of different shapes and depths such as blind holes, through-holes, and scratches can arise on the surface of the exploded foil transducer element due to factors such as holes, pits, burrs, foreign particulate matter, and target droplets on the substrate surface [11,12,13,14,15]. Eliminating film defects entirely is challenging [16], and even when films are deposited using HiPIMS, an advanced physical deposition method, a significant number of defects can still occur [17]. The generation of such defects can significantly impact the detonation time and transduction efficiency of explosive foil transducer elements.

Currently, there are limited theoretical studies focusing on the surface defects of bridge foils and their influence on the electrical explosive performance of exploding foil transducer elements. Most studies primarily rely on experimental methods [18]. In this study, through-hole defects are used as an example to investigate the effect of bridge defects on the electrical explosive performance of copper foil transducer elements through a combination of theoretical simulations and experimental methods. The theoretical simulations involve establishing a finite element analysis software-based model for calculating the temperature and current-voltage characteristics of the bridge foil with through-hole defects. The explosion time and energy utilization of three types of copper foils (defect-free, 20 μm radius through-hole defects, and 50 μm radius through-hole defects) are computed. Alumina ceramics are employed as the substrate to prepare defect-free copper foils, 20 μm radius through-hole defects, and 50 μm radius through-hole defects by magnetron sputtering and photolithography, respectively. The impact of through-hole defects on the electrical explosion performance of the bridge foil is analyzed by measuring the electrical explosion current and voltage curves using Rogowski coils and high-voltage probes.

## 2. Theories and Numerical Simulations

### 2.1. Bridge Foil Electro-Thermal Conversion Theory

This study utilizes COMSOL (https://cn.comsol.com/) multi-physics field finite element analysis software to conduct electro-thermal analysis of metal copper bridge foil. The analysis assumes that all electrical energy is converted into heat during the electrical explosion process and that the volume of the metal phase change remains constant. Furthermore, the physical properties, such as electrical conductivity and specific heat capacity, are assumed to vary with temperature, while the thermal conductivity of each material is considered constant. The heat source intensity within the domain Ω, *Q_e_* (*x*, *y*, *z*, *t*), is defined as the electrical loss. According to Joule’s law [19], *Q_e_* can be expressed as follows: (1)$$Q_{e}=J\cdot E,$$
where *J* is the current density and *E* is the electric field strength. According to the law of conservation of charge, a current continuity equation holds within domain Ω: (2)$$I=-\int_{t_{2}}^{t_{1}}\underset{\Omega }{∭}J(x,y,z,t)ndS=\frac{d\rho }{dt},$$
where *ρ* is the space charge number density, and *I* is the capacitor short-circuit discharge current within the blast circuit.

From Equation (2), the current density in the bridge foil can be determined by calculating the current value in the exploded circuit. This calculation considers Faraday’s law [19] and Maxwell–Ampere’s law [19], allowing the expression of the current density *J* and the electric field strength *E* as follows: (3)J=σE, E=−∇φ,
where *σ* is the temperature-dependent conductivity in S/m, and *φ* is the potential in V. Substituting Equation (3) into Equation (1), the heat source intensity of the electrically generated heat of the bridge foil can be obtained as follows: (4)$$Q_{e}=J\cdot E=\sigma {\left|\nabla \varphi \right|}^{2}.$$

According to the heat calculation formula, the temperature is related to the heat source by the following equation: (5)$$\rho C_{p}\frac{\partial T}{\partial t}=Q_{e}=\sigma {\left|\nabla \varphi \right|}^{2}.$$

Extracting the temperature information, it can be obtained that: (6)$$\frac{\partial T}{\partial t}=\frac{\sigma {\left|\nabla \varphi \right|}^{2}}{\rho C_{p}}=\frac{J^{2}}{\sigma \rho C_{p}}.$$

Equation (6) reveals that by acquiring the conductivity of the copper material, the density of the copper material, and the constant-pressure heat capacity of copper, calculating the current density of the electrical explosion using the current value from the short-circuit capacitance is feasible. This calculation enables the determination of the surface temperature of the bridge foil.

### 2.2. Bridge Foil Electro-Thermal Conversion Simulations

The structure and size of the bridge foil determine the ignition performance of the transducer element. The larger the central area of the bridge foil, the more energy is required for ignition, which will increase the difficulty of ignition. On the contrary, if the central area is too small, the manufacturing process is complex for production. In this study, a 1:1 3D model of the copper foil transducer element is employed [20,21,22,23]. The central area of the bridge foil is designed to have dimensions of 0.3 mm × 0.3 mm, an angle of 45°, and a thickness of 4 μm.

The modeling results are shown in Figure 1a. To facilitate the meshing of small through-hole structures, a triangular grid is utilized for the central area of the bridge foil. The through-hole is positioned at the origin of the coordinates, and geometric models of through-hole defective bridge foils and defect-free bridge foils with radii of 20 and 50 μm are established, as illustrated in Figure 1b–d, respectively.

The initiation system for copper foil is generally considered to be an RLC series circuit, as shown in Figure 2. The initial value for numerical simulation is the discharge current of the capacitor in the explosive circuit. In the electric explosion experiment, the initial step involves charging the capacitor using a power supply. Once the voltage of the capacitor aligns with the power supply voltage, it can be considered that the capacitor charging is complete. After receiving the detonation command, the detonation system initiates the closure of the high-voltage switch, triggering the discharge of the capacitor and resulting in the formation of a pulse current. This pulse current is responsible for inducing the electrical explosion of the copper foil [24]. In Figure 2, L1 and R1 are the self inductance and resistance of the circuit, respectively. The copper foil bridge can be successfully initiated when the charging voltage is 1100 V with a capacitance of 0.22 µF [22]. Therefore, an initial voltage of 1100 V and a capacitance of 0.22 µF were applied in the following simulations and experiments, resulting in a short-circuit current passing through the bridge foil from left to right.

The temperature distributions of the bridge area at 200 ns after energizing the bridges with different radius through-hole defects are computed by COMSOL, and the results are presented in Figure 3. These results indicate that through-hole defects in the bridge foil cause the electrical explosion temperature to be concentrated at the defect site. Moreover, the temperature is lower on the left and right sides of the through-hole defect, whereas it is higher on the top and bottom sides.

To further analyze the temperature variations, the average temperature within the central area (0.3 mm × 0.3 mm) of the bridge foil was calculated for the three copper foil transducer elements, and the results are displayed in Figure 4.

The findings demonstrate that the presence of through-hole defects leads to a higher temperature in the bridge foil compared with the defect-free bridge foil. This temperature increase is primarily due to the higher temperatures observed on the top and bottom sides of the through-hole defects. This is due to the reason that, when the current flows through the foil bridge from left to right, the current flow around the through-hole will be disturbed, the current density along the tangent direction of the current flow (the top and bottom sides of the through-hole) is much higher than that along the current flow direction (the left and right sides of the through-hole), resulting in higher temperatures on the top and bottom sides of the through-hole defects. Consequently, the average temperature in the bridge area rises significantly faster in the presence of through-hole defects compared with that in the case of the defect-free bridge foil.

### 2.3. Calculations of Bridge Foil Current and Voltage

According to Kirchhoff’s law [19], the equation for the detonation circuit current is:(7)$$L\frac{dI}{dt}+I[R_{0}+R(t)]+\frac{1}{C_{0}}\int_{0}^{t}Idt=U_{0}.$$

For a charging capacitor, the following relationship holds:(8)$$U_{0}=-\frac{Q_{0}}{C_{0}}.$$

Substituting Equation (8) into Equation (7), the differential equation for the current can be obtained as follows: (9)$$L\frac{dI}{dt}+I[R_{0}+R(t)]+\frac{1}{C_{0}}(\int_{0}^{t}Idt+Q_{0})=0.$$

Solving Equation (9) yields the expression for the variation of current with time as follows: (10)$$I=U_{0}{\left[\frac{1}{LC_{0}}-{\left(\frac{R_{0}+R(t)}{2L}\right)}^{2}L\right]}^{-1}e^{\left(-\frac{(R_{0}+R(t))t}{2L}\right)}\sin(\sqrt{\frac{1}{LC_{0}}-{\left(\frac{R_{0}+R(t)}{2L}\right)}^{2}}t).$$
where *U*_0_ refers to the detonation circuit voltage, *C*_0_ the detonation circuit charging capacitance, *L* the detonation circuit parasitic inductance value, *R*_0_ the charging circuit resistance value, and *R*(*t*) the bridge foil explosion process resistance value. Equation (10) expresses the current in the detonation circuit in terms of *R*_0_, *L*, *R*(*t*), *C*_0_, and *U*_0_ to determine the current. The conductivity of the bridge foil can be calculated by the conductivity formula [25]:(11)$$\sigma (T)=\left\{\begin{array}{c} \sigma_{0}(1+\alpha (T-T_{0}))\;\;\;T\leq 3000\;K\\ \frac{n_{e}}{n_{a}^{2}\gamma_{v}zT}\;\;\;3000\;K\leq T<8000\;K\\ (\frac{n_{e}e^{2}\tau }{m_{e}})A^{\alpha }(\frac{\mu }{kT})\;\;\;\;T\geq 8000\;K \end{array}\right..$$
where $\sigma_{0}$ is the conductivity of the bridge foil when the temperature is $T_{0}$, α is the variation coefficient of conductivity with temperature, $n_{e}$ is the electron density (cm^−3^), $n_{a}$ is the atomic density (cm^−3^), *z* is the ion equivalent charge, $\gamma_{v}=\alpha_{v}/c_{p}$ with the metal volume expansion coefficient $\alpha_{v}(K^{-1})$ and metal isobaric specific heat capacity $c_{p}(J/(g\cdot K))$, *e* is the electron charge, τ is the electron energy relaxation time, $m_{e}$ is the electron mass, μ is the chemical potential of metal, *k* is the Boltzmann constant, and $A^{\alpha }$ can be calculated by the Fermi–Dirac integral.

Substituting the temperature results calculated in Section 2.2 into Equation (11), the bridge foil resistance *R*(*t*) can be calculated from the following equation:(12)$$R(t)=\frac{l}{C_{s}}{\left[\sigma (T)\right]}^{-1}.$$
where *l* and $C_{s}$ are the length and cross-section area of the bridge foil, respectively.

The above relationships lead to the construction of the following system of equations:(13)$$\left\{\begin{array}{l} y_{1}{}^{\prime }=\frac{I^{2}(t)}{C_{s}{}^{2}}=\frac{1}{C_{s}{}^{2}}y_{3}{}^{2}\\ y_{2}{}^{\prime }=I(t)=y_{3}\\ y_{3}{}^{\prime }=\{U-y_{3}[R_{0}+R(t)]-\frac{y_{2}}{C}\}\times \frac{1}{L} \end{array}\right..$$

Equation (13) was solved using the fourth-order Longacurta algorithm to obtain the explosion process current *y*_3_ of the exploding foil. The calculated electrical explosion current–voltage curves for bridge foils with various sizes of through-hole defects are presented in Figure 5.

### 2.4. Analysis of Simulation Results

The electrical explosion time of the copper foil is defined as the peak point of the voltage curve. By analyzing the voltage curves, the electrical explosion times of copper foils with different sizes of through-hole defects are determined, as depicted in Figure 6.

The energy transfer efficiency is a crucial parameter for assessing the electrical explosive performance of the bridge foil. It is defined as the ratio of the energy stored in the bridge foil to the stored capacitive energy of the detonation circuit:(14)$$\eta =\frac{W}{P}=\frac{2\int_{0}^{t_{0}}u\cdot idt}{CU_{0}^{2}},$$
where *u* and *i* are the voltage and current during the electroburst of the copper foil, respectively, *C* is the value of the energy storage capacitance, *U*_0_ is the initial detonation circuit charging voltage, and *t*_0_ is the effective energy deposition time.

In the EFI ignition system, the flying sheet acceleration process typically takes approximately 200 ns [22]. Therefore, a time node of 200 ns after the electrical explosion point of the bridge foil can be used to distinguish the effective energy deposition. Within this 200 ns period following the electrical explosion, it is considered as an effective energy deposition time. Any data beyond 200 ns after the electrical explosion are not included in the calculation of the effective energy deposition on the bridge foil. The simulation results listed in Table 1 indicate that through-hole defects have a significant impact on the electrical explosion time and conversion efficiency of copper foil transduction elements. Specifically, larger through-hole defects result in earlier electrical explosion times of the bridge foil and lower conversion efficiency.

## 3. Experiment and Results

In this study, three types of copper foil transducer elements were fabricated: (1) a defect-free copper foil, (2) a copper foil with a 20 μm radius through-hole defect, and (3) a copper foil with a 50 μm radius through-hole defect. First, copper film with the thickness of 4 μm was coated on alumina ceramic substrate by magnetron sputtering technique; then, the copper film was etched into the butterfly shaped bridge shown in Figure 1 by photolithography technique; the through-hole defects were artificially manufactured on the bridge by laser drilling technique.

To measure the current and voltage during the electrical explosion, the Rogowski coil method were utilized [22]. The experimental system for the electrical explosion test of the exploding foil includes a high-voltage pulse power supply, a Rogowski coil, a Digital high-voltage meter, and an oscilloscope, as illustrated in Figure 7.

The high-voltage pulse power supply is an RLC series detonation circuit, as depicted in Figure 2, to ignite the bridge foils. The Rogowski coil is used to measure the current of the circuit, the digital high-voltage meter is used to measure the voltage of the circuit, and the oscilloscope is used to collect and display the current and voltage values measured by the Rogowski coil. The current and voltage curves obtained during the electrical explosion process of the three types of copper foil are presented in Figure 8.

The simulated calculations for the defect-free bridge foil, 20 μm radius through-hole defect bridge foil, and 50 μm radius through-hole defect bridge foil were compared with the experimentally measured explosion time, bridge foil deposition energy, and explosion current voltage. The comparison results are summarized in Table 2.

For the defect-free and 20 μm through-hole bridge foils, the experimental and calculated values are similar, with a minimum relative error of 0.44% and a maximum relative error of 3.44%; for the 50 μm through-hole bridge foil, the error between the experimental and calculated values increases with a maximum relative error of 12.06%. The results demonstrate suitable agreement of the simulation model with the experiment.

Compared with the defect-free bridge foil, when the through-hole defect radius is 20 μm, the bridge foil’s electrical explosion time is advanced by 1 ns, and the energy conversion efficiency is reduced by 1.52%. Similarly, when the through-hole defect radius is 50 μm, the bridge foil’s electrical explosion time is advanced by 51 ns, and the energy conversion efficiency is reduced by 13.96%.

## 4. Conclusions

In this study, the impact of through-hole defects on the electrical explosion time and transduction efficiency of bridge foils was investigated using a combination of finite element numerical simulations and experiments. The following three conclusions are drawn from the results:(1)Through-hole defects lead to an advancement in the explosion time, and the size of the through-hole determines the extent of this advancement. The relationship between the through-hole radius and the bridge foil explosion time follows a quadratic function.(2)Through-hole defects significantly reduce the transduction efficiency of the bridge foil transducer element during electrical explosion.(3)Specifically, when the through-hole radius is 20 μm, the through-hole defect causes the bridge foil explosion time to advance by 1 ns, and the energy conversion efficiency is reduced by 1.52%. When the through-hole radius increases to 50 μm, the through-hole defect leads to a 51 ns advancement in the bridge foil explosion time and a 13.96% reduction in the energy conversion efficiency. When the structure of the bridge area or the size of the through-hole changes, the values of the explosion time advance and the energy conversion definitely will vary; the specific values need to be further determined through simulations and experiments.

These findings highlight the critical role of through-hole defects in influencing the electrical explosion behavior and performance of bridge foils.

## Figures and Tables

**Figure 1 micromachines-14-01499-f001:**
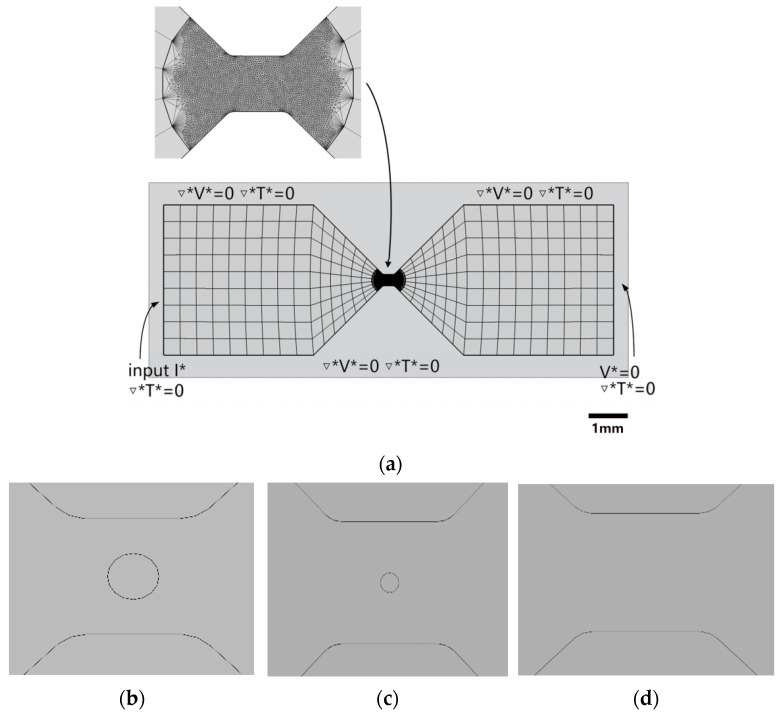
Modeling results of copper foil transducer: (**a**) Bridge foil meshing; (**b**) 50 μm radius through-hole bridge foil; (**c**) 20 μm radius through-hole bridge foil; and (**d**) defect-free bridge foil.

**Figure 2 micromachines-14-01499-f002:**
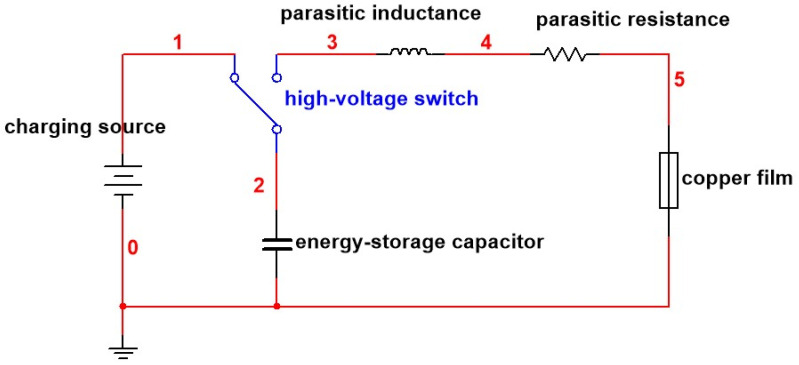
Initiation circuit for copper foil.

**Figure 3 micromachines-14-01499-f003:**
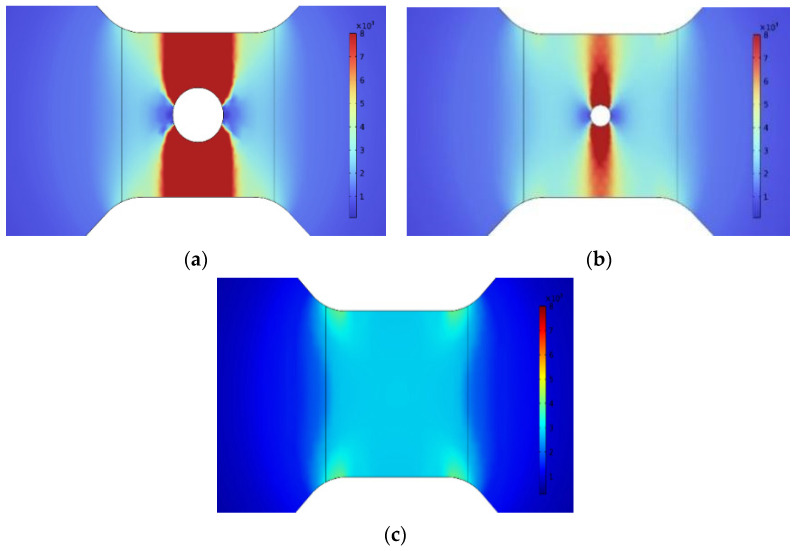
Calculated temperature gradient distribution at 200 ns for three types of bridge foil. (**a**) 50 μm radius through-hole bridge foil; (**b**) 20 μm radius through-hole bridge foil; and (**c**) defect-free bridge foil.

**Figure 4 micromachines-14-01499-f004:**
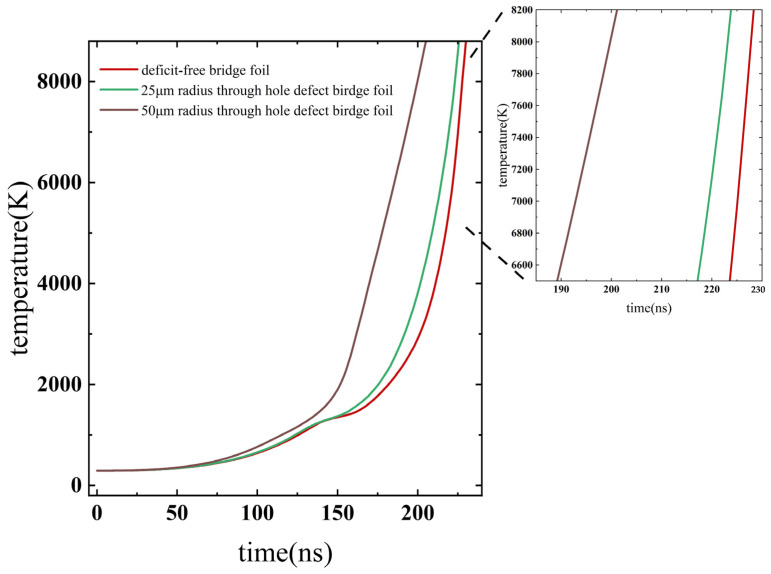
Effect of through-hole defects on the average temperature in the bridge area of the copper foil.

**Figure 5 micromachines-14-01499-f005:**
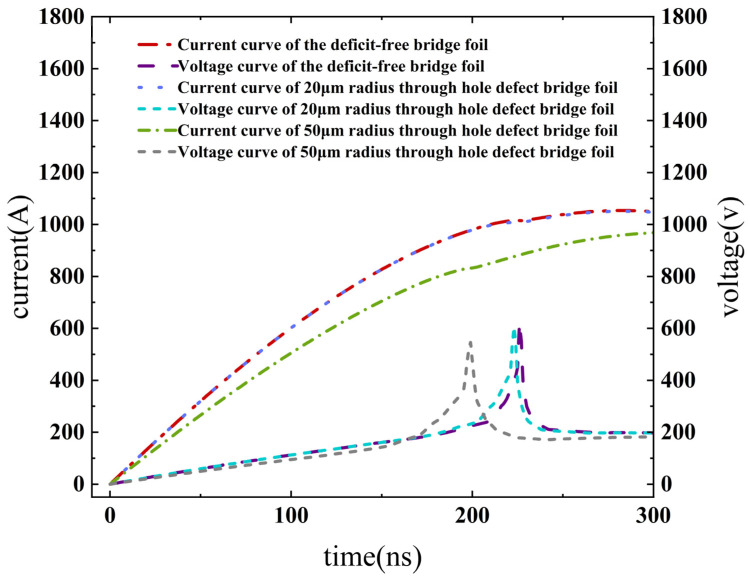
Copper foil electrical explosion current–voltage curve.

**Figure 6 micromachines-14-01499-f006:**
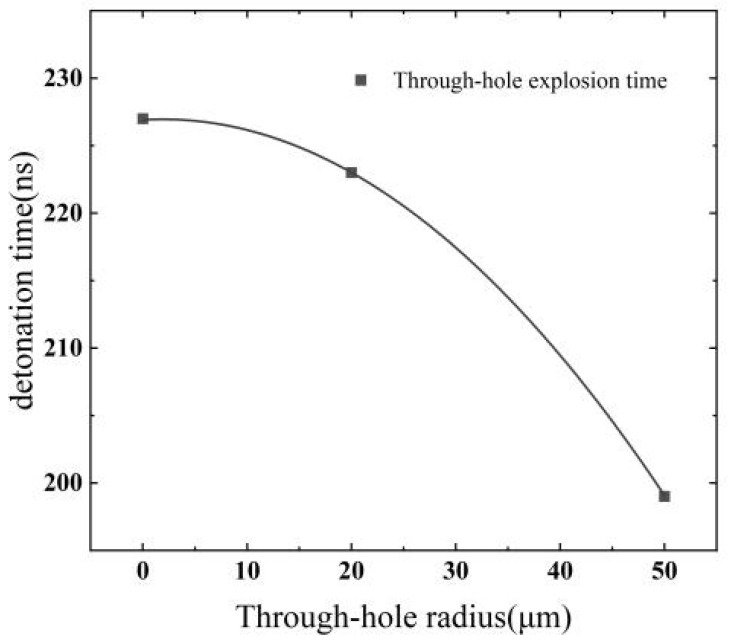
Electrical explosion time for copper foils with different defect sizes.

**Figure 7 micromachines-14-01499-f007:**
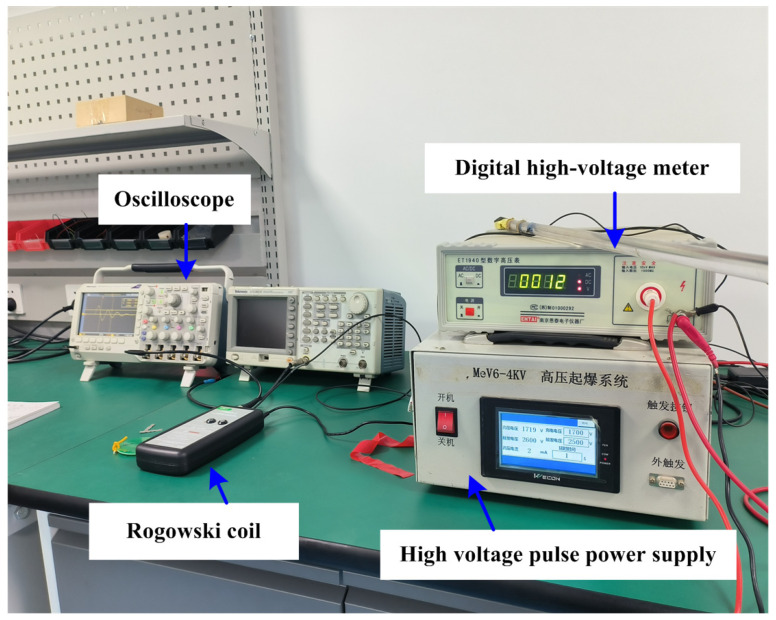
Experiment setup for ignition.

**Figure 8 micromachines-14-01499-f008:**
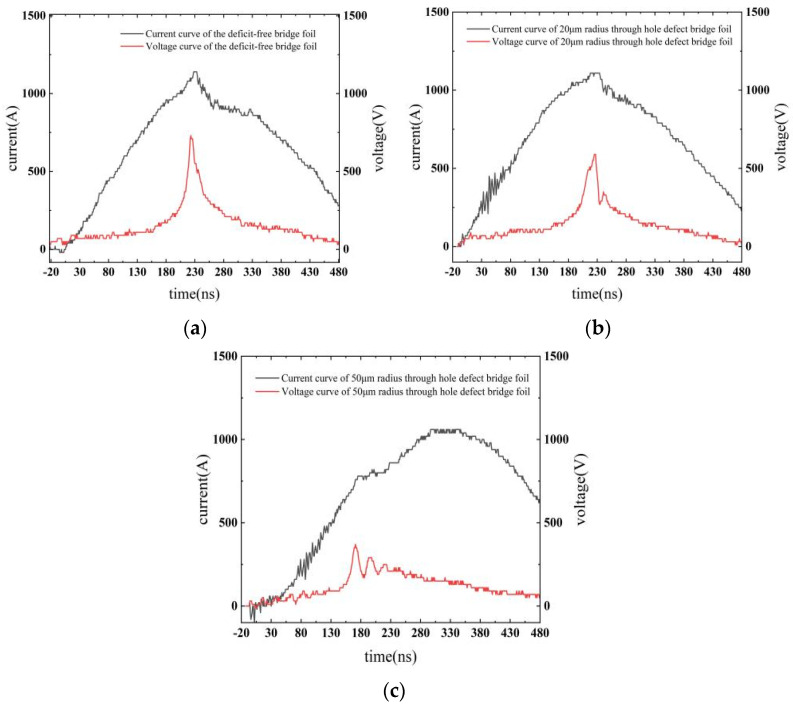
Experimentally measured current-voltage curves for the three bridge foils. (**a**) Defect-free bridge foil current-voltage curve; (**b**) Bridge foil current-voltage curve for a 25 μm radius through-hole defect; (**c**) Bridge foil current-voltage curve for a 50 μm radius through-hole defect.

**Table 1 micromachines-14-01499-t001:** Parameters of electro-explosive properties.

Defect Size	Explosion Time/ns	Deposited Energy/mJ	Capacitive Energy Storage/mJ
Defect-free bridge foil	227	62.86	133.10
20 μm radius through-hole defect	223	62.26	133.10
50 μm radius through-hole defect	199	49.39	133.10

**Table 2 micromachines-14-01499-t002:** Comparisons of the electro-explosive parameters by calculation and experiment.

Defect Size	Parameters	Experimental Values	Calculated Values	Relative Errors
defect-free bridge foil	Outbreak time/ns	226	227	0.44%
	Deposited energy/mJ	62.15	62.86	1.13%
	Energy conversion efficiency	46.69%	47.23%	1.14%
20 μm through-hole bridge foil	Outbreak time/ns	225	223	−0.90%
	Deposited energy/mJ	60.12	62.26	3.43%
	Energy conversion efficiency	45.17%	46.78%	3.44%
50 μm through-hole bridge foil	Outbreak time/ns	175	199	12.06%
	Deposited energy/mJ	43.56	49.39	11.80%
	Energy conversion efficiency	32.73%	37.11%	11.80%

## Data Availability

All data that support the findings of this study are included within the article.

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
