# Peer review of "Impact of Through-Hole Defects on the Electro-Explosive Properties of Exploding Foil Transducers"

_micromachines, 2023, doi:10.3390/mi14081499_

Round 1
Reviewer 1 Report
The manuscript studied the impact of through-hole defect on the electro-explosive properties of exploding foil transducer by numerical simulations and experiment. It is valuable for improving the electro-explosive efficiency of bridge foil elements. The physical models for electro-thermal conversion numerical simulation and current-voltage computation are reasonable, and the simulation results are well proved by experiment. I think the manuscript was well organized and written except some questions in the following:
1. The electric explosion characteristics of the transducer is a research hotspot. In the first part, the latest reference should be added and summarized.
2. In the Page 4, Figure 2 and 3 display different temperature gradient distribution caused by different holes, the reason should be explained in the manuscript.
3. In Eq. (11), what does“σ” mean?
4. The resolution of Figure 3 and 4 should be improved.
5. In figure 6. (b) experiment setup, the main device name should be clearly marked on the figure.
In short, I agree to accept the paper after a minor revise.
Reviewer 2 Report
Summary:
This paper presents a study focused on investigating the influence of surface defects on the electro-explosive properties of a specific metal explosive foil transducer. The research specifically examines the effects of defects in the bridge foil and their impact on the electrical explosion time and transduction efficiency. A simulation model is developed to analyze the behavior of a defective bridge foil, and the results are validated through experimental measurements.
The findings highlight that the presence of through-hole defects on the bridge foil's surface leads to an advancement in the electrical explosion time and a reduction in transduction efficiency. A comparison is made between the defective bridge foil and a defect-free copper foil, demonstrating the negative impact of surface defects on performance.
Overall, the paper is suitable for publication in the journal "Micromachines. However, improvements are needed in terms of providing a contextual background, describing the methodology in more detail, and discussing the practical implications. Addressing these areas would enhance the overall clarity and impact of the abstract, making it more informative for readers and researchers. I would recommend considering it for publication after major revision.
Major:
1. Generally, the language used in this paper is professional. The tone is analytical and focused on the content and structure of the abstract, providing constructive feedback to help improve the clarity and impact of the research.
2. In the experiment, the authors applied a specific design for the geometry of the bridge foil, which had dimensions of 0.3 mm × 0.3 mm, an angle of 45 degrees, and a thickness of 4 μm. However, it would be beneficial for the authors to provide further justification to illustrate why this design was chosen for the practical study.
3. Additionally, the authors describe the method by which the bridge area was energized, wherein an initial voltage of 1100 V and a capacitance of 0.22 μF were applied. As a result of this setup, a short-circuit current passes through the bridge foil from the left side to the right side. However, it would be helpful for the authors to provide further justification or reasoning for selecting these specific values of voltage and capacitance. Explaining why these particular parameters were chosen would provide insight into the experimental setup and its relevance to the study.
4. In the concluding session, the authors report that when the through-hole radius increases to 50 μm, it results in a 51 ns advancement in the bridge foil explosion time and a 13.96% reduction in the energy utilization rate. It is indeed important to consider the transferability of these findings to other cases since the experiment is focused on a specific area of 0.3 mm × 0.3 mm. If the observed effects are not universal and cannot be generalized to different dimensions or configurations, it would diminish the overall value and applicability of the research.
5. In line 161 on page 6, the authors mentioned that a polynomial fit was applied to the explosion time data, indicating a quadratic relationship between the explosion time and the through-hole radius. However, it is worth noting that the number of data points used for fitting the polynomial is relatively small, which raises concerns about the robustness and reliability of the research findings.
6. Regarding the experimental study, it is crucial for the authors to provide detailed information on how the copper foil was prepared. While the introduction briefly mentions the utilization of magnetron sputtering and photolithography techniques, it is essential to expand on these methods in the experimental section. Offering comprehensive explanations of these techniques would ensure the reproducibility of the research. Furthermore, it is essential for the authors to provide a thorough characterization of the materials used in the experiment. Specifically, presenting microscopic images of the through-hole foil would greatly contribute to the understanding and visualization of the surface defects being investigated.
Minor:
1. The authors provide a comprehensive explanation of the simulation process, incorporating various equations. While some of these equations may be considered basic, it would still be valuable to acknowledge and provide proper citations to the original sources to give credit to the experts in the field.
a) page 2 line 78: Joule’s law
b) Page 2 line 87: Faraday’s law
c) Page 2 line 87: Maxwell-Ampere’s law
d) Page 5 line 135: Kirchhoff’s law
2. It is worth noting that there are some notations used in the equations that require further elaboration and clarification to ensure a comprehensive understanding of their meaning and significance.
Page 5 equation 11, notations need to be expanded.
3. In Table 2, it would be more appropriate and professional to present the relative error instead of a simple error column to depict the difference between the experimental and laboratory results. Utilizing the relative error provides a standardized measure of the deviation between the two sets of data and allows for a more accurate comparison.
4. Figure 1 missing scale bar

The language used in the paper is professional. It employs formal and objective language to provide an evaluation of the abstract, highlighting its strengths, and areas for improvement, and offering an overall assessment. The tone is analytical and focused on the content and structure of the abstract, providing constructive feedback to help improve the clarity and impact of the research. For some specific sentences and paragraphs, there are opportunities for further refinement to elevate the overall quality of the paper.
Round 2
Reviewer 2 Report
The authors have provided proper justification for the manuscript, both in terms of language use and scientific aspects. I would highly recommend publishing this paper.
The language used in this paper is professional. The tone is analytical, providing a clear illustration of the research scope. I would recommend publishing this paper in present form